# Health Status Evaluation of Welding Robots Based on the Evidential Reasoning Rule

**Bang-Cheng Zhang** [1,2,*], **Ji-Dong Wang** [1], **Shuo Gao** [1], **Xiao-Jing Yin** [1] **and Zhi Gao** [1]

[1] The School of Mechanical and Electrical Engineering, Changchun University of Technology, Changchun 130103, China
[2] The School of Mechanical and Electrical Engineering, Changchun Institute of Technology, Changchun 130103, China
[*] Correspondence: zhangbangcheng@ccut.edu.cn

**Abstract:** It is extremely important to monitor the health status of welding robots for the safe and stable operation of a body-in-white (BIW) welding production line. In the actual production process, the robot degradation rate is slow and the effective data are poor, which can reflect a degradation state in the large amount of obtained monitoring data, which causes difficulties in health status evaluation. In order to realize the accurate evaluation of the health status of welding robots, this paper proposes a health status evaluation method based on the evidential reasoning (ER) rule, which reflects the health status of welding robots by using the running state data monitored in actual engineering and through the qualitative knowledge of experts, which makes up for the lack of effective data. In the ER rule evaluation model, the covariance matrix adaptive evolutionary strategy (CMA-ES) algorithm is used to optimize the initial parameters of the evaluation model, which improved the accuracy of health status evaluations. Finally, a BIW welding robot was taken as an example for verification. The results show that the proposed model is able to accurately estimate the health status of the welding robot by using the monitored degradation data.

**Keywords:** welding robot; health status evaluation; evidential reasoning rule; CMA-ES algorithm

## 1. Introduction

With the continuous progress of science and technology and the popularization of the intelligent manufacturing industry, the traditional automobile manufacturing industry is developing towards modern automation with robots as the main workforce. The automobile manufacturing industry is an important symbol of a country's industrialization degree and has a decisive influence on the national economy. Welding production, as one of the four major processes of the automobile, is an important process that determines the quality and performance of the automobile industry. The welding robot is a very important industrial robot in body-in-white (BIW) welding production lines, so the evaluation of its health status provides a significant guarantee for the safe and reliable operation of the whole production line. In the automobile welding production, a large number of industrial robots are used in a wide variety of stations. The safe and reliable operation of industrial robots plays a decisive role in the automobile welding process. The health status and performance of industrial robots affects the quality of welding products and can even lead to unqualified products. Thus, more and more enterprises pay more attention to the health management of industrial robots on the production line. For example, Xiao [1] proposed a health evaluation and status prediction algorithm based on the hidden Markov model (HMM) and the time convolutional network (TCN), aiming at the problems of high labor costs, low efficiency, and the low accuracy of mechanical shaft health management in industrial robot applications. A further study [2] used nuclear density estimation and Kullback–Leibler distance to detect deviations in torque repeatability of industrial robot

joints. Compared with healthy robot joints, degraded robot joints require higher torque to complete specific tasks.

The welding robot is a very important industrial robot on the BIW welding production line, so the evaluation of its health status provides a significant guarantee for the safe and reliable operation of the whole production line. The presence of a welding robot ensures the quality of welding production and, at the same time, prevents workers from directly engaging with the complex and dangerous welding environment. However, with the increase in usage time and the influence of various uncertainties, the degree of health and running status of the welding robot are more or less affected. Once the health status of a certain part of a welding robot deteriorates, the equipment can fail or even cause more serious disasters, resulting in accidents involving human personnel. Therefore, it is of great significance to evaluate the health status of BIW welding robots, allowing experts and maintenance personnel to carry out timely and appropriate maintenance and repair on the welding robots, in accordance with the evaluation results.

In order to evaluate the state of a welding robot accurately, it is necessary to establish an effective evaluation model. At present, the mainstream health status evaluation models of complex electromechanical systems are divided into three types:

(a) Model-based methods, which are predominantly applied in systems where the working mechanisms and fault mechanisms can be established. Additionally, they can better identify and evaluate the state of electromechanical equipment by constructing mathematical expressions that describe the degradation process of the performance, such as the Kalman filter [3], analytic hierarchy process [4], hidden Markov model [5], and Bayesian network methods [6]. Although this kind of method has achieved good results in engineering, it is difficult to establish an accurate model for complex electromechanical systems with many variables and influencing factors in the operation stage.

(b) Data-driven methods, which generally use a large amount of current or historical monitoring data to build nonlinear models of the system, such as the support vector machine [7,8], convolutional neural network [9,10], decision-making tree [11], and random forest methods [12]. Such methods do not need to establish accurate mathematical models and have strong ability to simulate realistic systems but often need a large amount of data to train models. For complex electromechanical systems such as welding robots, it is difficult to monitor a large amount of effective data, and many data-driven methods belong to the "black box" model, which cannot be interpreted. So, this type of method is not suitable to evaluate the health status of complex electromechanical systems.

(c) Qualitative knowledge-based methods, which mainly use experience and domain knowledge, such as the expert system [13], fault tree [14], and Petri net methods [15]. These methods have the ability to deal with uncertain information, and reasoning processes are highly explanatory and require very little data. However, due to the complexity of complex electromechanical system and the limitation of expert knowledge, the accuracy of evaluation is reduced. In addition, in actual production, the monitoring operation data can most intuitively reflect the health status of the equipment at this time. Therefore, when evaluating the health status of a complex electromechanical system, it is necessary to use both qualitative knowledge and appropriate data to accurately evaluate the state of the complex electromechanical system.

Although the above three mainstream evaluation methods can achieve acceptable results, the BIW welding robot belongs to the complex production process of multi-working procedures and multi-working conditions. As a result, it is particularly difficult to establish an accurate mathematical analytical model, as it is an environment characterized by complex structure, strong correlation, uncertainty, and high nonlinearity. Therefore, accurate evaluation results cannot be obtained solely by relying on qualitative knowledge, and data-driven methods lack interpretability. Moreover, the above three methods do not have the ability to deal with uncertain information. The data-driven approach lacks interpretability, and the above three methods lack the ability to deal with many forms of uncertain information. However, evaluation methods based on semi-quantitative information could

well solve the limitations of the above methods, of which the evidential reasoning (ER) rule method proposed by Yang and Xu [16] in 2013 is able to effectively utilize quantitative monitoring data and qualitative expert knowledge to assess the health status of complex electromechanical systems, such as welding robots. The ER rule is a multiple criteria decision analysis (MCDA) problem proposed on the basis of D-S evidence theory and decision theory. The ER rule method makes up for the traditional MCDA method by establishing a unified confidence frame to describe many types of uncertainties in multi-attribute decision making problems.

It is unique in that it fully considers the weight and reliability of each piece of evidence, which enables it to deal with ambiguity, uncertainty, and incompleteness. Moreover, qualitative knowledge can be effectively used in the ER rule method, which makes up for the shortcomings of welding robots, which lack effective data. Based on this, this paper puts forward an evaluation model of the welding robot's health status based on the ER rule, and its results are interpretable, which also solves the difficulties of poor evaluation results caused by relying solely on qualitative knowledge and quantitative data. Additionally, the ER rule assessment model has been widely used in laser gyro [17], aerospace systems [18], and aerospace relay [19].

In the absence of a large number of effective monitoring data, in order to make better use of qualitative knowledge to evaluate the welding robot state evaluation, this paper proposes a welding robot health status evaluation model based on the ER rule. The rest of this article is as follows. The Section 2 describes the health status evaluation of welding robot. The Section 3 introduces the health status evaluation of the BIW welding robot based on the ER rule. The Section 4 verifies the effectiveness of the ER rule evaluation model in practical cases. The Section 5 is the discussion of this paper. The Section 6 summarizes this paper.

## 2. Problem Description of the ER Rule-Based Health Assessment Model

At present, a large number of welding robots are used in the automatic welding production lines of the automobile industry, and their health status affects the welding quality of the automobile body, so it is of great engineering value to evaluate the health status of welding robots.

Based on the analysis of the working mechanism and health mechanism of the BIW welding robot, this section selects the indicators that are able to reflect the health status of the welding robot, such as reliability indicators, failure indicators, performance indicators, and running status indicators. Of the many indicators, the running status indicator, which can accurately reflect the health status of the welding robot, is also a key monitoring indicator of enterprises. Today, more and more enterprises export the internal running status data of robots through the robot data acquisition network box to provide data support for the health management of welding robots. In the running status data, torque, speed, and working angle are key concerns of businesses, and the monitoring data can also directly reflect the state of welding robots. Therefore, starting from real-world engineering contexts, this paper takes the running status index as an example and carries out the health status assessment of the welding robot according to the running status data of a certain model of welding robot provided by an enterprise. Figure 1 shows the health indicator system of this paper.

The main process of the health status evaluation of a BIW welding robot is divided into four parts: (a) data acquisition, (b) data preprocessing, (c) the determination of health indicators, and (d) the construction of the curve of quantitative value of health status. The following are the specific steps for the health status evaluation of BIW welding robots:

Step 1: The dataset through the data of a certain welding robot provided by the enterprise is obtained.

Step 2: By analyzing the health mechanism of the BIW welding robot, the index which accurately reflects the state of the welding robot is selected as the input of the evaluation model.

Step 3: The data of the input of the evaluation model are pre-processed, including noise reduction and normalization.

Step 4: In order to evaluate the performance of the evaluation model, monotonicity index (*Mon*) [20] and correlation index (*Corr*) [21] are selected as objective functions, and the covariance matrix adaptive evolutionary strategy (CMA-ES) algorithm is used to search the optimal weight parameters corresponding to the global maximum of monotonicity index and correlation index.

Step 5: The utility value of the time series is calculated by evaluating the model, so as to obtain the quantitative value of the health status of the welding robot, and the curve is connected according to the time series. Figure 2 is the flow chart of the health status assessment model.

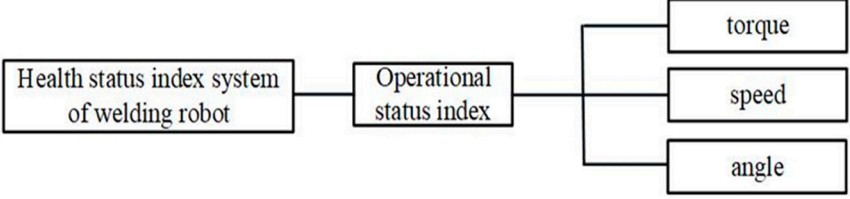

**Figure 1.** Operating condition index of a BIW welding robot.

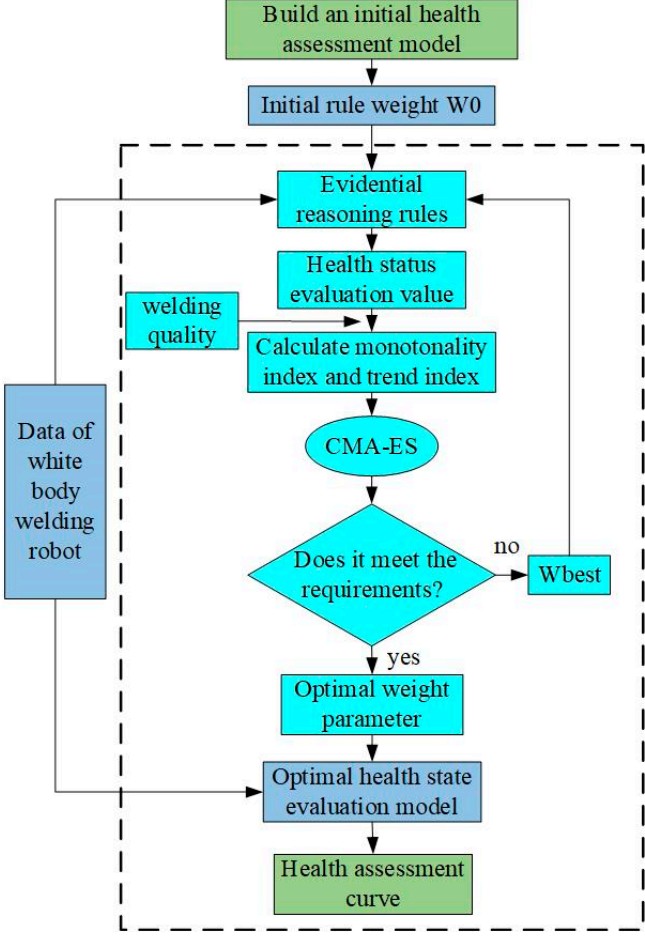

**Figure 2.** Flow chart of the health status assessment model.

## 3. Health Status Evaluation of a BIW Welding Robot Based on the ER Rule

### 3.1. ER Rule

The inference rule is a further inference and extension of D-S evidence theory, which belongs to a MCDA problem. Reasoning rules use the same confidence rule framework to solve the uncertain problems in multi-attribute decision-making problems, which is a

further supplement to traditional MCDA methods, such as the analytic hierarchy process (AHP), and solves its limitations and rationality when dealing with uncertain information. In this section, the ER rule is used to construct the performance degradation evaluation model of the BIW welding robot.

Before using the ER rule for evidence fusion, it is necessary to process the quantitative data and convert them into the confidence distribution of the relative reference level. It is assumed that the $n^{th}$ reference value of evidence (also known as attribute $x_i$) process can be represented as $h_{n,i}(n = 1, 2, \ldots, N)$ and $h_{n+1,i} \geq h_{n,i}$, where $h_{n,i}$ represents the maximum reference value and $h_{n,i}(n = 1, 2, \ldots, N)$ represents the minimum reference value. When the initial reference value of the evidence is obtained, the quantitative data can be converted into a confidence distribution, namely:

$$S(x_i) = \{(h_{n,i}, \beta_{n,i}), n = 1, 2, \ldots, N\} \tag{1}$$

where $\beta_{n,i} = \dfrac{h_{n+1,i} - x_i}{h_{n+1,i} - h_{n,i}}$; $\beta_{n+1,i} = 1 - \beta_{n,i}$; $h_{n,i} \leq x_i \leq h_{n+1,i}$; $\beta_{m,i} = 0$, $m = 1, 2, \ldots, N$, $m \neq n, n + 1$.

The following distribution can be used to express quantitative attributes:

$$S(x_i) = \{(H_{n,i}, \beta_{n,i}), n = 1, 2, \ldots, N\} \tag{2}$$

where $\beta_{n,i} \geq 0$, $\sum\limits_{n=1}^{N} \beta_{n,i} \leq 1$, $\beta_{n,i}$ represents the confidence that the quantitative attribute $x_i$ is evaluated as reference level $H_{n,i}$. $S(x_i)$ represents the confidence distribution of quantitative attribute $x_i$ relative to the reference level. When attribute $x_i$ is single, it is assumed that $m_{n,i}$ represents the basic reliability value that the individual is evaluated as reference level $H_n$, and $m_{H,i}$ represents the basic reliability value that is not assigned to any reference level $H_n$, namely:

$$m_{n,i} = \omega_i \beta_{n,i}, n = 1, 2, \ldots, N \tag{3}$$

where $\omega_i$ is the weight of quantitative attribute $x_i$ and $0 \leq \omega \leq 1$, $\sum\limits_{i=1}^{L} \omega_i = 1$.

$$m_{H,i} = 1 - \sum_{n=1}^{N} m_{n,i} = 1 - \omega_i \sum_{n=1}^{N} \beta_{n,i} \tag{4}$$

In strict the ER rule method, $m_{H,i}$ can be decomposed into two parts, namely, $\widetilde{m}_{H,i}$ and $\overline{m}_{H,i}$, and:

$$\widetilde{m}_{H,i} = \omega_i \left(1 - \sum_{n=1}^{N} \beta_{n,i}\right) \tag{5}$$

$$\overline{m}_{H,i} = 1 - \omega_i \tag{6}$$

where $\widetilde{m}_{H,i}$ indicates the incomplete degree of evaluation of attribute $x_i$, and when the confidence distribution is complete, that is, when the unallocated confidence is 0, $\widetilde{m}_{H,i} = 0$. $\overline{m}_{H,i}$ represents the residual basic reliability value.

If the $i^{th}$ attribute and the $(i+1)^{th}$ attribute are aggregated using the ER rule method, then the reliability distribution value of the evaluation level is $H_n$. Then, the reliability distribution values at the evaluation level are as follows:

$$\{H_n\}: \ m_{n,I(i+1)} = K_{I(i+1)} \left[m_{n,I(i)} m_{n,i+1} + m_{H,I(i)} m_{n,i+1} + m_{H,i+1} m_{n,I(i)}\right] \tag{7}$$

$$m_{H,I(i)} = \overline{m}_{H,I(i)} + \widetilde{m}_{H,I(i)} \tag{8}$$

$$\{H\}: \widetilde{m}_{H,I(i+1)} = K_{I(i+1)}\left[\widetilde{m}_{H,I(i)}\widetilde{m}_{H,i+1} + \widetilde{m}_{H,I(i)}\overline{m}_{H,i+1} + \overline{m}_{H,i+1}\widetilde{m}_{H,I(i)}\right] \tag{9}$$

$$\{H\}: \overline{m}_{H,I(i+1)} = K_{I(i+1)}\left[\overline{m}_{H,I(i)}\overline{m}_{H,i+1}\right] \tag{10}$$

$$K_{I(i+1)} = \left[1 - \sum_{t=1}^{N}\sum_{\substack{l=1 \\ l \neq t}}^{N} m_{l,i+1}m_{t,I(i)}\right]^{-1}, \quad (i = 1,\ldots,L-1) \tag{11}$$

When all attributes are synthesized according to the iterative synthesis algorithm, the confidence is allocated as follows, namely:

$$\{H\}: \hat{\beta}_n = \frac{m_{n,I(L)}}{1 - \overline{m}_{H,I(L)}}, \quad (n = 1,\ldots,N) \tag{12}$$

$$\{H\}: \hat{\beta}_H = \frac{\widetilde{m}_{H,I(L)}}{1 - \overline{m}_{H,I(L)}} \tag{13}$$

In the above formula, $\hat{\beta}_n$ represents the confidence that the attribute is evaluated as the reference level $H_n$, and $\hat{\beta}_H$ represents the confidence assigned to the global level. The confidence range can be expressed as $[\beta_n, (\beta_n + \beta_H)]$.

The overall evaluation $y$ can be expressed as the following confidence distribution, which is as follows:

$$S(y) = \{(H_n, \hat{\beta}_n), n = 1,\ldots,N\} \tag{14}$$

Then, the influence degree of the input data on the performance state of the BIW welding robot is calculated by using the following formula:

$$X(y) = \sum_{n=1}^{N} H_n \times \hat{\beta}_n \tag{15}$$

Of these, $N$ is the grade number of evaluation results and $X(y)$ is the utility value of individual $y$ after the iterative fusion of $L$ attributes.

### 3.2. The Objective Function of the Model

Combined with the research of relevant experts and scholars at home and abroad, this section takes monotonicity index and correlation index as the optimization objective function to optimize the health status evaluation model of welding robots.

(a)　Monotonicity index

In actual production and life, the degradation process of each component of a welding robot is irreversible, that is, the degradation characteristics should show a monotonic decreasing or increasing trend, known as monotonicity. The monotonicity index formula is:

$$Mon(F_i) = \left|\frac{\text{Num of } dF_i > 0}{T_i - 1} - \frac{\text{Num of } dF_i < 0}{T_i - 1}\right| \tag{16}$$

where $dF_i$ represents the difference of feature sequence values. Num of $dF_i > 0$ and Num of $dF_i < 0$ represent the number of positive and negative values in the feature sequence, and $T_i$ represents the total number of features.

(b)　Correlation index

With the increase in service time, the health of a welding robot may gradually deteriorate, which is called correlation. The correlation index indicates the correlation between degradation trend and time. The formula of the correlation index is:

$$Corr(y_t) = \frac{\left| \sum\limits_{t=1}^{T_t} (y_t - \bar{y})\left(l_t - \bar{l}\right) \right|}{\sqrt{\sum\limits_{t=1}^{T_t} (y_t - \bar{y})^2 \sum\limits_{t=1}^{T_t} \left(l_t - \bar{l}\right)^2}} \tag{17}$$

where $\bar{y}$ represents the average value of the quantified values of health status, $l_t$ is the number of the $t^{th}$ sampling point, and $\bar{l}$ is the average value of the number of sampling points.

The range of *Mon* and *Corr* is 0 to 1. The higher the *Mon* value, the better the monotone of the model, the faster the convergence rate, and the higher the evaluation efficiency. The larger the *Corr* value, the higher the correlation degree between the model and the real data and the more accurately the results can be reflected. Therefore, the objective function of the parameter optimization of the ER rule model established in this paper is as follows:

$$\begin{aligned}
& \max\{Mon(\Omega)\} \\
& \max\{Corr(\Omega)\} \\
& \text{s.t. } 0 \le \tilde{w}_i \le 1, \ i = 1, 2, \cdots, T \\
& \Omega = \{\tilde{w}_1, \tilde{w}_2, \cdots, \tilde{w}_T\}
\end{aligned} \tag{18}$$

### 3.3. CMA-ES Optimization Algorithm

In the ER rule evaluation model, the health status is regarded as a process of dynamic degradation, which can be described by certain characteristics. First, key characteristics affecting health status are selected, and then expert knowledge is used to build an initial evaluation model. The initial parameters of the ER rule evaluation model are given by experts and may not be accurate. Therefore, the CMA-ES algorithm [22–24] is adopted in this paper to optimize the initial parameters of the evaluation model. When solving complex nonlinear non-convex optimization problems in a continuous domain, CMA-ES has no gradient optimization, does not use gradient information, and can converge to the global optimal point in a relatively short time with fewer individuals, and is thus the most advanced algorithm in evolutionary computing. There are two objective functions of this model, namely, the monotone coefficient index and the time correlation coefficient index. First, all equality conditions are converted into objective functions, and it should be noted that multiple objective functions can be established for the solutions. Then, these objective functions are independent of each other, so the CMA-ES algorithm can be used to solve them separately. The implementation details of CMA-ES are as follows:

(a) Set initial values:

Taking a solution in the solution space (the space formed by parameter vectors $\Omega^0$ of the ER rule model) as the center point, the initial population is generated with a normal distribution. The initial average $m^0 = \Omega^0$, initial covariance matrix $C^0$, initial step size $\sigma$, and overall size $\lambda$ are obtained.

(b) Generate initial population:

The $\Omega^0$ is selected as the expectation. The method of generating the population is as follows:

$$\Omega_k^{g+1} \sim m^g + \sigma^g \mathbb{N}(0, C^g), \ for \ k = 1, \ldots, \lambda \tag{19}$$

where $\Omega_k^{g+1}$ represents the $i^{th}$ solution of the generation $(g+1)^{th}$, $m$ represents the overall mean, $\sigma$ represents the step size, $\mathbb{N}$ represents normal distribution, and $C^g$ represents the covariance matrix of generation $g^{th}$.

(c)　Select operation:

This operation is used to select the top $\varepsilon$ optimal solutions in the population based on the value of fitness function $f(\Omega)$, and the smaller the value, the better the solution.

(d)　Reorganization operation:

This operation is used to update the expectation of the population and cause it to shift towards the optimal solution, so as to guide the evolution of the population towards the optimal solution during the process of regenerating the population. The method of updating the population expectation is as follows:

$$mean^{g+1} = \sum_{i=1}^{\varepsilon} \gamma_i \Omega_{i:\lambda}^{g+1} \tag{20}$$

where $\gamma_i$ denotes the weight of the $i^{th}$ solution, and its sum should be equal to 1. $\lambda$ represents the number of solutions in the population. $\Omega_{i:\lambda}^{g+1}$ represents the $i^{th}$ solution of the $g^{th}$ generation in $\lambda$ solutions.

(e)　Update $C$ operation:

The $C$ represents the elliptic plane of equal probability density of the population distribution. The initial matrix $C^0$ is the identity matrix $I$, and the corresponding population distribution equaling the probability density surface is the unit sphere. The $C$ is updated according to the following equation:

$$C^{g+1} = (1 - \alpha_1 - \alpha_\varepsilon)C^g + \alpha_1 q^{g+1} \left(q^{g+1}\right)^{\mathrm{T}}$$
$$+\alpha_\varepsilon \sum_{i=1}^{\varepsilon} \gamma_i \left( \frac{\left(\Omega_{1:\lambda}^{g+1} - mean^g\right)}{\eta^g} \right) \left( \frac{\left(\Omega_{1:\lambda}^{g+1} - mean^g\right)}{\eta^g} \right)^{\mathrm{T}} \tag{21}$$

where $\alpha_1$ and $\alpha_\varepsilon$ denote the learning factor. $q$ denotes the evolution path, and the initial evolution path value is 0. The rule is updated below:

$$q^{g+1} = \left(1 - \alpha_q\right)q^g + \sqrt{\alpha_q\left(2 - \alpha_q\right)\left(\sum_{i=1}^{\varepsilon} \gamma_i^2\right)^{-1}} \frac{mean^{g+1} - mean^g}{\eta^g} \tag{22}$$

where $\alpha_q \leq 1$ denotes the evolution path parameter. The step size $\eta$ is updatedbelow:

$$\eta^{g+1} = \eta^g \exp\left( \frac{\alpha_\eta}{d_\eta} \left( \frac{\left\|q_\eta^{g+1}\right\|}{E\|\mathbb{N}(0,I)\|} - 1 \right) \right) \tag{23}$$

where $d_\eta$ denotes the damping coefficient, $E\|\mathbb{N}(0,I)\|$ denotes the expectation of Euclidean paradigm $\mathbb{N}(0,I)$. $I$ denotes the identity matrix. $q_\eta$ denotes the conjugate evolution path. $\alpha_\eta$ denotes the conjugate evolution path parameter. $q_\eta$ is updated below:

$$q_\eta^{g+1} = \left(1 - \alpha_\eta\right)q_\eta^g + \sqrt{\alpha_\eta\left(2 - \alpha_\eta\right)\left(\sum_{i=1}^{\varepsilon} \gamma_i^2\right)^{-1}} C^{(g)-\frac{1}{2}} \frac{mean^{g+1} - mean^g}{\eta^g} \tag{24}$$

The above operation steps should be repeated until the accuracy requirements meet output of the optimal parameter $\Omega_{best}$ of the ER rule model.

## 4. Case Studies

In order to verify the effectiveness of the above method, this section adopts the monitoring data of ABB welding robot irb-6700 model from the front floor working group of

an automobile welding production line of a specific model of motor vehicle for verification. The data are collected through the robot controller and exported by the robot data acquisition network box installed by the business. The data are collected in as a time series, with a sampling frequency of 50 hz and a sampling time of 640 s. A total of 32,000 data samples were collected. Since the monitored degradation data were very slow and there were no large fluctuations over long periods of time, this paper will process and extract the degraded data of the collected time series, divide it into 100 groups of data on average, and then extract the root mean square of each group of data, and finally obtain the experimental data, as shown in Figure 3 below. These data include the torque (a), speed (b), and operating angle (c) of the welding robot.

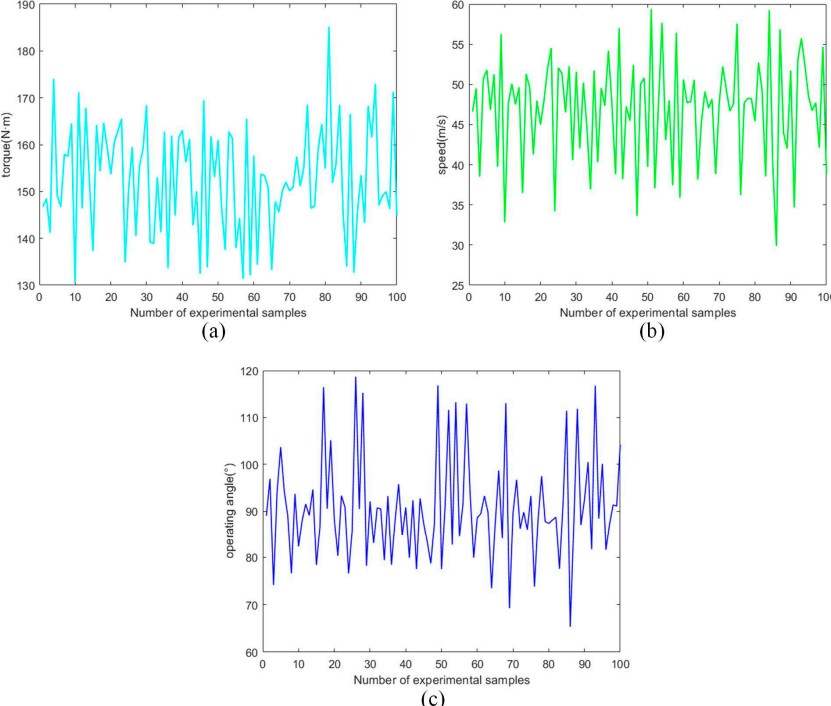

**Figure 3.** Experimental data.

The above data are input into the established optimized ER rule evaluation model, and the results are shown in Figure 4. As can be seen from Figure 4, the health status of welding robot slowly deteriorates with the increase in time of usage, which is consistent with the problems in actual engineering. When the degradation reaches a certain threshold value, the staff of the enterprise need to undertake maintenance of the robot.

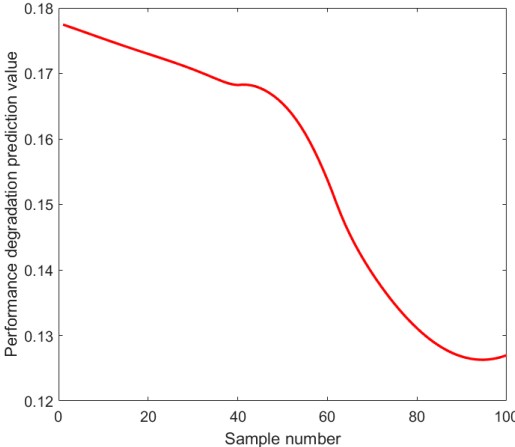

**Figure 4.** Evaluation results of optimized ER rule model.

In order to verify the effectiveness of the model, we compared it with other models. The comparison models included traditional the ER rule (parameters not optimized), BP neural network, convolution neural network, and fuzzy C-means methods. The comparative experimental results are shown in Figure 5. All methods used the same set of data samples. See Steps 1–5 in Section 3 for the modeling process of the ER rule method. This model provides the same initial weight parameters as the traditional ER rule model, and these parameters are determined by expert experience.

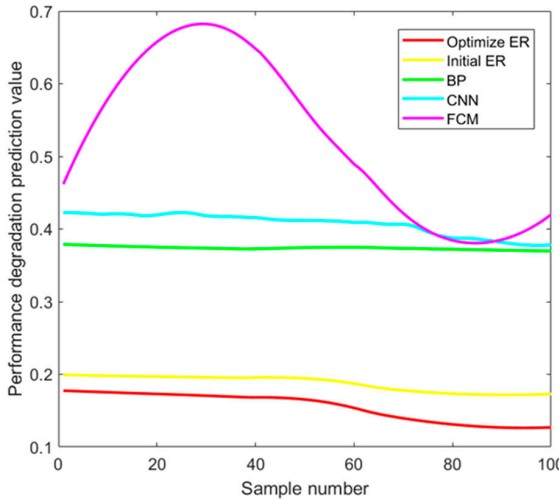

**Figure 5.** Comparative experimental results.

The results of specific performance evaluation index values are shown in Table 1 below. It should be noted that the greater the monotonicity coefficient, the stronger the upward or downward trend of the performance degradation curve. The larger the correlation coefficient, the stronger the correlation between the performance degradation curve and the real data.

**Table 1.** Experimental results of different methods.

|  | Optimized ER | BP | CNN | FCM | Initial ER |
|---|---|---|---|---|---|
| Monotonicity coefficient | 0.8788 | 0.5960 | 0.6566 | 0.1313 | 0.7172 |
| Correlation coefficient | 0.9584 | 0.8594 | 0.9311 | 0.7779 | 0.9393 |

The trend of the performance degradation curve of the model is shown in Figure 5. With the increase in the number of samples, i.e., an increase in time, the performance degradation curve of the welding robot decreases slightly and the value of the performance degradation factor decreases, indicating that the robot is slowly undergoing performance degradation. Table 1 shows the monotonicity coefficient and correlation coefficient of the performance degradation curve, among which the two indexes of this model are closer to 1, that is, the objective function value is the largest, indicating that the model proposed in this paper can effectively evaluate the health status of a BIW welding robot.

It can be seen from the above results that the ER rule model proposed in this paper is able to accurately evaluate the state of a welding robot with fewer data samples.

## 5. Discussion

The results show that the ER rule has obvious advantages in the health status assessment of welding robots by using expert knowledge. In the health status assessment modeling of this type of complex mechanical and electrical equipment, the degradation process of equipment is slow and the monitoring data are relatively stable, which means that it is difficult to sensitively reflect changes in health status due to the lack of effective data. In the modeling process of this kind of problem, industry expert knowledge is used

to provide correct trend information of status change in order to make up for the lack of effective monitoring data. If the effective monitoring data are abundant, other data-driven modeling methods will also have a good effect on the health status assessment of welding robots.

## 6. Conclusions

The welding robot is one of the key pieces of equipment in the welding production line of BIW. The stable and reliable operation of welding robots is very important for the safe and reliable operation of the welding production line and a guarantee of the welding quality of BIW. In order to make better use of quantitative data and expert knowledge in order to realize the nonlinear health evaluation model of welding robots, a health evaluation model based on the ER rule was proposed. It can be seen from the experimental results that the ER rule evaluation model has achieved optimal results in the two objective function indicators, indicating that the model can obtain better results under relatively small samples. In order to obtain the optimal parameters of the ER rule model, the CMA-ES algorithm is used for parameter optimization, and the weight corresponding to the maximum value of the objective function is the optimal weight of the model. The experimental results show that, with continued use, the performance of a welding robot will gradually degenerate, which is consistent with real-world engineering contexts. In addition, the model is compared with other models. The results show that the model is expected to provide a new method for the health status evaluation of welding robots and has broad application prospects.

The model can also be applied to other complex electromechanical systems, such as aircraft engines. In future studies, how to improve the accuracy of expert knowledge in the process of ER rule modeling in combination with the characteristics of the object or how to establish a health status assessment model that is closer to engineering practices are problems that we should continue to study.

**Author Contributions:** Conceptualization, B.-C.Z. and S.G.; methodology, J.-D.W. and S.G.; software, J.-D.W.; validation, B.-C.Z., J.-D.W. and S.G.; formal analysis, X.-J.Y.; investigation, Z.G.; resources, B.-C.Z.; data curation, Z.G.; writing—original draft preparation, J.-D.W. and S.G.; writing—review and editing, X.-J.Y.; visualization, Z.G.; supervision, S.G.; project administration, B.-C.Z.; funding acquisition, B.-C.Z. All authors have read and agreed to the published version of the manuscript.

**Funding:** The research was supported by the Jilin Provincial Science and Technology Development Project under grants 20210301033GX, 20200401114GX, and 20200301038RQ.

**Data Availability Statement:** The data has not been made public because of concerns about corporate privacy.

**Conflicts of Interest:** The authors declare no conflict of interest.

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
