# Peer review of "Health Status Evaluation of Welding Robots Based on the Evidential Reasoning Rule"

_electronics, doi:10.3390/electronics12081755_

Round 1

Reviewer 1 Report

The topic of a paper titled “Health status evaluation of welding robot based on evidential reasoning rules” is important and interesting, especially from the presented system reliability perspective. I have some comments to improve the paper:

1. Figure 1 – I don’t understand the title of this figure, what does it mean: “This is a figure. Schemes follow the same formatting”?

2. Lines 107-108 – something is missing: in brackets “In order to evaluate the performance of the evaluation model, monotonicity 107 index ( ) and trend index ( ) are selected as objective functions”.

3. The discussion paragraph is missing and the summary is too general and short.

4. It is not explicitly specified which extremum in the objective function is looking for.

5. Has the correct operation of the presented model been verified and the coefficients validated or how it was done?

I have also a few suggestions regarding text formatting in the future:

1. The authors use the same numbering style in the text and for formulas in the form of (x). eg. lines 41, 46, and for all formulas.

2. Most numbering of formulas is incorrect and formatting is illegible.

3. The charts in Figure 3 are too small and illegible.

4. Lines 241-242 and table 1 – something is wrong in formatting.

5. There are places where spaces are missing, e.g. line 45.

6. The paper requires significant editorial corrections.

I think the research topic is important and has potential but at the moment paper requires significant editorial corrections and some substantive corrections. I also want to know if the presented model can be easily adapted for other systems, not only for welding robots, and what changes will it require. What are the further scientific plans of the authors regarding the presented topic?

Author Response

word.  Please see the attachment.

Reviewer 2 Report

Dear authors, thank you for your effort, after thorough reads and revision, it can be understand that the purpose of this article/study is to propose the ER rules to evaluate the health condition of BIW. Prior presenting the results of the data, authors MUST follow the AUTHOR GUIDELINES carefully and thoroughly in order to present the manuscript draft, the systematic arrangement of the manuscript, citation required, attachment of materials and methods to assure the reproducibility and another details. Also, the materials and methods is one of the essential passage in research article to provide information required by the potential reader/audience. Results and discussion is also presented in a less-logical and comprehensive fashion, which in turn cause the confusion. Eventually, after a careful peer-reviewing process, the decision toward this manuscript will be rejection. In addition, the detail comments can be found in the attached file.

Thank you, 

Sincerely

Author Response

word.  Please see the attachment.

Round 2

Reviewer 1 Report

The authors of the paper took into account most of the comments. In my opinion, the form can be accepted for further processing (some text editing errors need corrections).

Author Response

word
